# Structure and Function of Multimeric G-Quadruplexes

**DOI:** 10.3390/molecules24173074

**Published:** 2019-08-24

**Authors:** Sofia Kolesnikova, Edward A. Curtis

**Affiliations:** 1The Institute of Organic Chemistry and Biochemistry of the Czech Academy of Sciences, 166 10 Prague, Czech Republic; 2Department of Biochemistry and Microbiology, University of Chemistry and Technology, 166 28 Prague, Czech Republic

**Keywords:** G-quadruplex, dimer, tetramer, multimer, oligomer, telomere, promoter, R-loop, DNA:RNA hybrid

## Abstract

G-quadruplexes are noncanonical nucleic acid structures formed from stacked guanine tetrads. They are frequently used as building blocks and functional elements in fields such as synthetic biology and also thought to play widespread biological roles. G-quadruplexes are often studied as monomers, but can also form a variety of higher-order structures. This increases the structural and functional diversity of G-quadruplexes, and recent evidence suggests that it could also be biologically important. In this review, we describe the types of multimeric topologies adopted by G-quadruplexes and highlight what is known about their sequence requirements. We also summarize the limited information available about potential biological roles of multimeric G-quadruplexes and suggest new approaches that could facilitate future studies of these structures.

## 1. Introduction

The B-form double helix is the most well known nucleic acid structure, but it is not the only one. Other examples include double helices with geometries that differ from that of classical B-form DNA, such as Z-DNA [1], triple helices [2], and even four-stranded structures in which canonical A-T and C-G base pairs are absent [3,4]. Among these noncanonical folds, the G-quadruplex (GQ) has received the most attention. This is a four-stranded structure typically stabilized by stacked guanine tetrads connected by short loops [3,5,6]. Because of their high stability, structural versatility, and functional diversity, GQs have been widely used as building blocks and functional elements in fields such as synthetic biology [7,8]. Sequences with the potential to form GQs are also abundant in the genomes of higher eukaryotes [9,10], and recent studies using GQ-specific antibodies indicate that these structures can form in the context of cells [11,12]. GQs are thought to play a variety of biological roles. These including regulation of transcription, translation, DNA replication, and RNA localization [13,14,15].

Most biological studies to date have focused on monomeric GQs. However, GQs can also adopt a variety of multimeric forms. These include relatively small structures such as dimers. They also include larger ones like G-wires, which can contain hundreds of GQ monomers. Although the ability of GQs to multimerize has long been recognized, the possibility that such high-order structures form in the context of cells has received less attention. From our perspective, however, two observations suggest that this possibility should be considered. First, the cellular concentrations of GQs are surprisingly high, especially in higher eukaryotes. For example, current estimates suggest that human cells contain at least 716,000 DNA GQs [16] in a volume (for a HeLa cell nucleus) of 0.22 pL [17]. This corresponds to a cellular GQ concentration of 6 μM. Not all of these GQs will be present at the same time in the cell cycle, and not all will be capable of forming dimers. On the other hand, this concentration is orders of magnitude higher than that required for efficient GQ multimerization. For example, in a recent study we identified GQs with dissociation constants of dimer formation as low as 35 nM [18]. The hypothesis that GQs form multimeric structures in cells is also compelling when the myriad evolutionary and functional advantages of this mechanism are considered [19,20,21,22]. These include the ability to regulate biochemical function on the basis of concentration, to detect ligands with enhanced sensitivity by cooperative binding, and to modulate activity in a rapid and reversible manner by the exchange of dimerization partners (Figure 1). Inspired by these considerations, in this review we describe the types of multimeric topologies adopted by GQs and review what is known about their sequence requirements. We also summarize the limited information currently available about the potential biological roles of multimeric GQs in cells and suggest new approaches that could facilitate future studies of these structures, especially in the context of cells.

## 2. Definition of Multimeric G-Quadruplexes and Modes of Multimerization

A multimer is an aggregate of molecules consisting of multiple monomers. Nucleic acids typically multimerize (hybridize) through duplex formation via the Watson–Crick base pairing of two complementary strands. A double helix provides DNA with structural stability, determined by hydrogen bonding and base-stacking interactions, and facilitates replication [23,24], while double-stranded RNA facilitates genetic interference [25]. Duplex formation is the most common mechanism of multimerization, but not the only one. Multimerization can also occur using a distinct hydrogen-bonding pattern called Hoogsteen base pairing. This enables formation of a G-tetrad, the building block of a unique type of nucleic acid structure called a G-quadruplex (GQ). To form a G-tetrad, four guanines associate via eight hydrogen bonds from both the Watson–Crick and Hoogsteen faces of the base (Figure 2).

G-tetrads stack on top of one another giving rise to a GQ (Figure 3). The types of sequences that can adopt GQ structures in vitro have been extensively characterized. To form a GQ, a sequence typically needs to contain segments of at least two guanines separated by mixed-sequence loop nucleotides. The most widely used models allow loops of one to seven nucleotides [9,10] but in some cases loops can be longer [26,27,28]. In addition, it is becoming increasingly clear that GQs can accommodate bulges [29], and noncanonical tetrads have also been observed in high-resolution structures [30,31,32,33,34,35,36]. Even more complicated topologies are seen in the Spinach, Mango, and Class V GTP aptamers, in which the four clusters of guanines that form the tetrads in the GQ are far apart in the primary sequence [37,38,39]. Guanines in the G-tetrad can in principle come from one, two, three, or four guanine-rich (G-rich) strands. GQs formed from only one strand are typically defined as intramolecular (unimolecular) (Figure 3A–B), although, as discussed below, multiple GQs on a single strand can also interact to form higher-order structures. GQs that contain more than one strand are termed intermolecular (multimolecular or multimeric) and can be classified according to the number of strands as bimolecular (dimeric) (Figure 3C), trimolecular (trimeric) (Figure 3D), or tetramolecular (tetrameric) (Figure 3E). Multimeric GQs formed from identical strands are called homomultimeric, whereas those composed from nonidentical strands are called heteromultimeric.

While the number of molecules in a GQ is the standard way to classify multimers, another approach is to consider the structure of the multimerization interface. GQs utilize two primary modes of multimerization. In the first mode, interfaces are formed by tetrads from two different GQs stacked on top of one another. When intramolecular (unimolecular) GQs multimerize using this mechanism, individual nucleic acid strands first assemble into monomeric GQs, such that all four guanines in each G-tetrad come from the same nucleic acid strand. Monomeric GQ subunits then stack on top of one another via π–π stacking interactions of terminal interfaces to form higher-order GQ structures (Figure 4A). Intermolecular (multimeric) GQs can also multimerize in this way if all of the guanines in one of the tetrads at the interface come from one GQ and all of the guanines in the other come from a different GQ (Figure 4A). GQ subunits can stack in three different orientations: 5′ to 3′ (head-to-tail) [40], 5′ to 5′ (head-to-head) [41], and 3′ to 3′ (tail-to-tail) [42]. Experimental and molecular dynamics data suggest that 5′ to 5′ stacking is the most common orientation for GQ structures due to a favorable stacking geometry [41,43]. The propensity of subunits to stack is also influenced by the topology of intermolecular GQ subunits. Stacking is favorable for parallel GQs (i.e., GQs with all strands oriented in the same direction and with propeller loops on the sides of the tetrads). In antiparallel GQ structures, which have strands in opposite orientations, lateral and diagonal loops are positioned above and below the GQ axis, which can impede stacking interactions of terminal G-tetrads [44]. Stacking interactions can also facilitate formation of higher-order structures from tandem GQ subunits folded on a single strand. Such a structure was proposed as one of the models of the human telomere overhang [45,46]. In addition to playing important roles in multimerization, stacking interactions are the most important mode of ligand binding to GQs. End-stacking GQ ligands typically bind to terminal GQs tetrads, but can also intercalate between tandem GQs to stabilize the multimeric structure [47,48,49,50,51,52].

In the second mode of multimerization, guanines from two or more nucleic acid strands hydrogen bond to form tetrads, so that interfaces occur within rather than between tetrads. These intermolecular G-tetrads then stack to form GQs of various lengths (Figure 4B), sometimes using slipped strands [53]. Various G-rich oligonucleotides multimerize via this mode, and a large body of literature has investigated their formation [5,6]. As early as 1988, Sen and Gilbert demonstrated that oligonucleotides containing motifs of four, five, or six contiguous guanines fold into tetrameric structures [54]. One year later, two different groups proposed the formation of dimeric GQs from two telomere-derived G-rich sequences [55,56]. Sen and Gilbert [57] observed that short oligonucleotides with three consecutive guanines at the 3′ end assembled into four-, eight-, and twelve-stranded GQ structures. Formation of these structures starts with the self-assembly of slipped tetramers, in which strands are not perfectly aligned but contain two overhanging guanines at the 3′end. Tetrameric subunits associate with one another via hydrogen bonding between these extra guanines.

Higher-order GQs which combine these two modes of multimerization have also been described [18,42,58,59,60,61]. An illustrative example is a GQ structure assembled from eight d(TGGGGT) strands (Figure 4C). The structural subunit is a tetramolecular GQ consisting of four perfectly aligned strands, each in an identical 5′–3′ orientation (second mode of multimerization). Two structural subunits then stack at their 5′ interfaces, forming an octamer (first mode of multimerization) [58,59].

Higher-order GQ structures are typically initially characterized using low-resolution techniques such as circular dichroism, dimethylsulfate footprinting, native PAGE, mass spectrometry, and analytical ultracentrifugation. These methods can be used to establish that the sequence forms a GQ, identify the guanines in tetrads, and determine the number of strands in the structure. Higher-order GQs can be visualized in greater detail using NMR and X-ray crystallography. Examples of high-resolution structures which utilize the first mode of multimerization include 5′–5′ stacked dimers with canonical [41,62,63] or extended [64] tetrads at the interface. Structures which use the second mode of multimerization include interlocked dimers which occur in the promoters and introns of oncogenes [65,66,67]. A structure which combines these two modes of multimerization is that of a parallel-stranded tetrameric GQ, formed from d(TGGGGT) strands [58,59]. Additional examples are discussed in [5,50,68] and elsewhere.

## 3. Sequence Requirements of Multimeric G-Quadruplexes

The propensity of a GQ sequence to fold into a particular multimeric structure in vitro depends on a number of factors, including the type and concentration of cation in the buffer, the presence of molecular crowding agents, the nucleic acid concentration, the sequence length, and the sequence composition [5,68,69,70]. Under conditions approximating the intracellular environment, the main factor controlling higher-order structure formation is the sequence. In the context of GQ formation, a large number of sequences, ranging from short strands with only one G-run to up to 20,000 nucleotide long sequences with roughly 3300 G-runs, have been investigated [58,59,71]. The key sequence features affecting multimerization are discussed below and summarized in Figure 5.

The minimum requirement for a sequence to fold into an intramolecular GQ is typically four runs of at least two guanine bases separated by loops ranging from one to seven nucleotides (Figure 5A). Sequences falling into this category often either fold into monomeric GQs that do not interact with one another, or form multimeric structures by the first mode of multimerization (stacking of monomeric subunits). Structure prediction is nevertheless complicated by reports of sequences containing four stretches of guanines which form intertwined dimers using the second mode of multimerization [65,66,67,72]. Sequences with fewer than four G-runs can typically only adopt GQ structures by interacting to form multimeric structures using the second mode of multimerization (Figure 5B). Sequences with two or three G-runs separated by short loops usually combine both types of multimerization and readily assemble into structures with eight, twelve, or even more strands [57]. At the extreme end of this continuum lie the G-wires [73], long linear ladder-like structures formed from numerous slipped GQ tetrameric subunits (Figure 6). G-wires are longer than any other higher-order GQs, with maximum lengths depending on the method of preparation as well as the sequence. For example, a DNA sample of d[G_4_T_2_G_4_] formed linear G-wires ranging from 7 to 100 nm in length. A 100 nm G-wire was calculated to contain 75 GQ blocks composed of 140 full and 20 half strands [74]. Positioning of the GQ subunits within the structure can be controlled by adding GC bases to the terminal ends, which enables the formation of G:C:G:C tetrads linking the subunits [75].

In addition to the number of G-runs in the sequence, multimerization also depends on the length of the G-run (Figure 5C). An instructive example is a study of structural transitions caused by truncations of guanine tracts in the telomere-derived DNA strand d(G_4_T_4_G_4_) [76]. The original sequence and its 5′ truncated analog d(G_3_T_4_G_4_) formed stable dimeric structures that did not undergo conversion into higher-order structures. In contrast, the 3′ truncated sequences d(G_3_T_4_G_3_) and d(G_4_T_4_G_3_) formed a mixture of dimeric, trimeric, and tetrameric GQs in a concentration-dependent manner. This structural polymorphism can be explained by varying stabilities of the dimeric structures formed by the reference and the various truncated sequences. The reference d(G_4_T_4_G_4_)_2_ dimer contains sixteen guanines which form four G-tetrads, and is therefore more stable than the three-tetrad dimers formed by truncated sequences. A bimolecular structure formed from two d(G_3_T_4_G_4_) strands is special in that, in contrast to the antiparallel dimers formed by d(G_3_T_4_G_3_) and d(G_4_T_4_G_3_), it contains three parallel strands. This specific strand orientation stabilizes the d(G_3_T_4_G_4_)_2_ dimer and thereby prevents it from rearranging into higher-order structures.

A third feature of G-runs that can affect multimerization is the presence of bulges (interruptions of G-runs by non-G nucleotides) (Figure 5D). Despite bulges being tolerated by various GQs, they can greatly influence structural stability. The extent to which GQ stability is reduced depends on a number of factors, such as the location of the bulges within the GQ sequence, the context of the sequence, and the overall GQ topology [29]. The introduction of bulges does not necessarily result in structural changes, and in some cases they do not prevent monomeric GQs from forming. On the other hand, GQs with reduced stability are prone to structural reorganizations, including higher-order structure formation. Bulged nucleotides as drivers of multimerization were systematically investigated in a recent study that analyzed dimerization and tetramerization as a function of guanine substitutions in G-runs [18]. The reference sequence used in this study, d(G_3_TG_3_AAG_3_TG_3_A), folds into a parallel three-tetrad GQ, with nucleotides 2, 6, 11, and 15 forming the central tetrad (Figure 7A). Despite containing an exposed 5′ tetrad, only monomers were observed when versions of this sequence which contained either a 5′ hydroxyl or a 5′ phosphate group were analyzed on native gels. Dimerization and tetramerization were induced by introducing substitutions of guanines for other nucleotides at certain positions in the central tetrad. Bulged G-runs prevented these sequences from forming stable intramolecular GQs by themselves and increased their propensity to form multimeric structures. Sequences containing substitutions at position 2, 6, or both were proposed to form intertwined dimers with a core of three canonical G-tetrads at the 3′ end (Figure 7B). Sequences with substitutions at positions 11, 15, or both were instead proposed to form intertwined dimers containing a core of three canonical G-tetrads at the 5′ end (Figure 7C). Such sequences also formed tetramers which, in accord with the most favorable mode of stacking, were proposed to consist of two dimers stacked in a 5′ to 5′ orientation (Figure 7C). Intertwined dimers formed from sequences with other possible combinations of substitutions (2 and 11, 2 and 15, 6 and 11, 6 and 15) would only contain two stacked G-tetrads [18]. This structural arrangement cannot effectively stabilize the structures, which probably explains why these sequences did not also multimerize.

GQ sequences can be designed to begin and/or end with G-runs or they can contain overhanging non-G nucleotides at the 5′ and/or 3′ ends (Figure 5E). Flanking nucleotides often inhibit stacking interactions by sterically hindering formation of interfaces by terminal tetrads [57,80]. In some cases even the introduction of a 5′ phosphate group is sufficient to inhibit 5′–5′ stacking [73]. This knowledge can be used when probing multimerization mechanisms: for multimers stacked in a 5′ to 5′ orientation, addition of flanking nucleotides at the 5′ end should interfere with the stacking interaction, while addition of flanking nucleotides to the 3′ end should not [18]. Nevertheless, some studies report the stacking of GQ subunits which contain overhanging nucleotides. In these structures, flanking nucleotides either radiate out from the GQ axis or become a part of various unusual structures such as thymine triads [81], guanine–uridine octads [82] or guanine–cytidine octads [60].

The role of loop nucleotides in multimeric GQ assembly has received relatively little attention from researchers, and most studies investigating multimeric GQs have not analyzed the effects of loop length and sequence composition. Despite this, examples in which loop nucleotides affect GQ multimerization have been described. Loop length can indirectly influence multimerization by affecting GQ topology (Figure 5F). Sequences bearing at least one single-nucleotide loop tend to adopt a parallel-stranded orientation, whereas longer loops increase the likelihood that sequences will fold into mixed-strand or antiparallel GQs [83]. As discussed above, this can affect the ability of GQs to interact by the first mode of multimerization: the propeller loops of parallel strand GQs do not generally affect their ability to form multimers through stacking on their terminal tetrads, whereas the lateral and diagonal loops of antiparallel GQs sometimes do. Loop sequence can also play an important role in GQ structure formation. One mechanism by which this can impact the global structure of the GQ is by formation of noncanonical extended G-tetrads. For example, d(GGA)_4_ strands assemble into dimers by stacking on a guanine–adenine heptad interface formed by two monomeric subunits. Three loop adenines provide hydrogen bonds donors and acceptors in the heptad plane and hence play an indispensable role in dimerization [64]. Moreover, several studies have shown that mutating nucleotides not thought to directly interact with tetrads can lead to major changes in the GQ structure. For example, changing the loops in a human telomere-derived sequence from TTA to AAA abolishes GQ formation [84]. Recent findings in our lab also show that mutations in loops can affect the propensity of sequences to form multimeric GQs [77,78].

In summary, these studies indicate that the main sequence features affecting multimerization are the number and length of the G-runs, the sequence composition of the G-runs, the presence of flanking nucleotides at the 5′ and 3′ ends of the GQ, and the length and sequence of loop nucleotides (Figure 5). Despite much being understood about the sequence requirements of multimeric GQs, it is generally not possible to predict higher-order GQ topology from sequence. Instead, experimental data from low-resolution methods such as circular dichroism, dimethylsulfate footprinting, native PAGE, mass spectrometry, and analytical ultracentrifugation are used in combination with high-resolution techniques such as NMR and X-ray crystallography.

## 4. Potential Biological Roles of Multimeric G-Quadruplexes at the Tips of Telomeres

The first discussions of biologically relevant higher-order GQ structures emerged as the length and sequence composition of telomeres was being characterized [85]. In the majority of eukaryotes, the nucleotide sequence of telomeres consists of a G-rich strand running 5′ to 3′ toward the end of the chromosome and a complementary C-rich strand. For most of their length, telomeres are double-stranded, except for a single-stranded G-rich 3′ overhang. While the telomeric sequence is similar in diverse species, the length of the overhang is not. For example, in *Tetrahymena* the overhang typically consists of tandem repeats of the sequence d(TTGGGG) and is 14 to 21 nucleotides long [86]. On the other hand, in humans the sequence of the overhang is d(TTAGGG) and it ranges in length from 125 to 275 nucleotides [87,88].

While the composition and length of the telomeric overhang has been characterized in many species, its secondary structure is still a matter of debate. The first studies of short telomere-derived synthetic oligonucleotides reported that these sequences folded into several topologically different guanine-rich structures [54,55,56,89]. Subsequent studies showed that these correspond to GQ structures, although several different topologies have been reported [6,90,91]. Examples include parallel-stranded tetrameric and antiparallel dimeric GQs assembled from short telomeric sequences bearing one and two G-runs, respectively, as well as intramolecular monomeric GQs formed by sequences containing four G-runs.

Because the human telomeric overhang contains multiple repeats of segments with four G-runs, it can potentially fold into a higher-order structure containing a series of monomeric GQs [92]. A single-stranded DNA with several monomeric GQs can adopt at least two different structural arrangements (Figure 8). The first one is termed “beads-on-a-string”, and assumes that monomeric GQ units do not interact with one another [93]. A second model proposes that GQ subunits associate with one another through stacking interactions between the terminal tetrads of consecutive GQs to form high-order structures [45,46]. One such arrangement of stacked monomers has recently been described in the context of nontelomeric oligonucleotides containing multiple GQs [94,95]. Despite evidence for both models, a structure in which GQs stack on one another appears to be most likely [92]. Structures consistent with this model have been observed by atomic force and electron microscopy [71,96,97], and methods such as FRET have provided additional evidence for interactions between neighboring GQ subunits [97]. Molecular dynamics simulations also support the idea that telomeric GQ monomers can stack to form higher-order structures [45,46]. On the basis of comparisons between calculated and experimentally measured sedimentation coefficients, it has been proposed that a structure consisting of hybrid rather than all-parallel GQs is most likely [98]. Despite these advances, a high-resolution structure of the telomeric overhang has not yet been reported.

In vivo evidence for the presence of GQs at telomeres came from studies reporting the binding of GQ-specific antibodies to the tips of telomeres [11,12,99]. These experiments also showed that telomerase co-localizes with GQ antibodies at chromosomal ends. Telomerase catalyzes telomere elongation and is directly involved in GQ unfolding [100]. Although these studies indicate that telomeric DNA adopts a GQ structure in vivo, they do not distinguish between monomeric and multimeric GQs. While antibodies specific for multimeric GQs have not yet been developed, several small molecules specific for multimeric GQs have recently been reported [51,101,102,103]. One of these compounds, IZNP-1 (triaryl-substituted imidazole derivative), binds and stabilizes multimeric GQs by intercalating into the cavity between two stacked GQ monomers. IZNP-1-treated cells contain a greater number of BG4 foci (human GQ structure-specific antibody foci) at telomeres than untreated cells. They also exhibit signs of DNA damage and telomere shortening, which can lead to cell cycle arrest, apoptosis, and senescence. Taken together, these findings suggest that multimeric GQs can form at telomeres under certain conditions and contribute to telomere dysfunction [51].

## 5. Potential Biological Roles of Multimeric G-Quadruplexes in Promoters

Visualization of IZNP-1-induced BG4 foci confirmed that telomeric ends have the ability to form multimeric GQs. Surprisingly, however, the percentage of fluorescent foci at telomeres was relatively low (38%), and the majority of antibodies localized to other genomic regions. One implication of this observation is the possibility that multimeric GQs exist outside telomeric DNA. If so, multimeric GQs might have roles beyond chromosomal capping.

A role for monomeric GQ structures in transcriptional regulation was proposed following the discovery that putative GQ sequences are highly enriched in nuclease hypersensitive regions within promoters [13]. In addition, more than 40% of human genes contain at least one GQ motif in their promoter. While formation of monomeric intramolecular GQs has been investigated for a number of these genes including *c-MYC, KRAS*, and *BCL2* [104,105,106,107], only a few examples of genes likely to be regulated by multimeric GQs in their promoters have been identified so far. The most well-characterized example is an unusual tetrad:heptad:heptad:tetrad (T:H:H:T) dimer, generated from a d(GGA) repeat sequence. An oligonucleotide containing four such repeats folds into an intramolecular structure in which a guanine tetrad and a guanine–adenine heptad are stacked on top of one another. The NMR structure indicates that two monomeric subunits stack 5′ to 5′ on the heptad plane, resulting in a dimeric T:H:H:T structure [64] (Figure 9). A similar structure was adopted by an oligonucleotide containing eight copies of the repeat [108]. The d(GGA) repeat sequence occurs in twelve nearly perfect tandem repeats in the *c-MYB* promoter. In theory, this sequence can give rise to three independent T:H building subunits (with four tandem repeats per building block), which could then stack in different combinations to form dimers. A study investigating this possibility proposed that a T:H:H:T motif formed by stacking interactions between the first and the third building blocks functions as a negative regulator of *c-MYB* promoter activity. The authors also identified a transcription factor, MAZ, which binds to the T:H:H:T structure and could play a role in the regulation of *c-MYB* expression [109,110]. The *c-MYB* promoter is not the only genomic region rich in GGA repeats, and similar motifs have been identified in the regulatory regions of other genes such as *NCAM* [111], *SPARC* [112], *KRAS* [113], and *CCNB1IP1* [114]. Although it is tempting to assume that the transcription of genes with multiple d(GGA) repeats in the promoter is regulated by T:H:H:T structures, the results presented in [109] and [110] do not completely rule out the possibility that inhibition of transcription is mediated by monomeric forms of these GQs. Testing the effects of antibodies or small molecules specific for the T:H:H:T structure on the extent of transcriptional inhibition could provide further insight into the structural organization of the d(GGA) repeat region in this promoter.

Another multimeric GQ structure that could potentially regulate gene expression occurs in the promoter of *hTERT*, the gene encoding the reverse transcriptase of telomerase. The core promoter of *hTERT* contains twelve consecutive G-runs and can therefore form multiple GQs. One proposed structural model for this region consists of two stacked GQs separated by a 26 nucleotide loop which contains another four G-runs and adopts a hairpin structure [115]. An alternative model suggests that all of the G-runs in the sequence participate in GQ folding and that the final structure consists of three parallel GQs tightly stacked on top of one another [40,116]. Although the available evidence does not indicate which model is correct, it does support the idea that a G-rich structure in this promoter regulates *hTERT* expression [115,116,117,118].

The hypothesis that multimeric GQs in promoters regulate gene expression is also supported by a study investigating the regulatory sequences of muscle-specific genes [119]. The promoter regions of these genes contain a high frequency of G-runs that form both homodimeric and heterodimeric structures in vitro. One example is the *ITGA7* promoter region, which contains two G-rich sequences, each capable of forming an intramolecular monomeric GQ, separated from one another by 85 nucleotides. These two G-rich sequences formed a heterodimeric structure when analyzed on native gels. This structure is hypothesized to play a role in the regulation of mouse myogenic gene expression. Consistent with this hypothesis, the myogenic determination protein MyoD binds the heterodimeric structure but not a monomeric GQ formed by one of the heterodimer-forming sequences. MyoD exhibited a similar binding preference for a homodimeric GQ formed by a G-rich region in the promoter of the *sMtCK* gene. Although this G-rich sequence occurs only once in the *sMtCK* promoter region and therefore cannot assemble into a homodimer in vivo, several G-rich clusters are present in adjacent genomic regions. It would therefore be of interest to determine whether a heterodimeric GQ structure can form in this region and is bound by MyoD [119].

Taken together, these data indicate that multimeric GQ structures can form in the promoter regions of several genes and are consistent with the idea that such structures can regulate gene expression. Many additional promoters contain multiple G-runs. For example, genes containing more than eight G-tracts include *TRIM13* [67], *c-MYC* [105,106], and *c-KIT* [120,121]. In addition, several viral promoters containing multiple sequences with the potential to form GQs have recently been reported [122,123]. Although these are candidates for genes regulated by multimeric GQs, our current understanding of GQ folding precludes reliable prediction of higher-order GQ structure from sequence. As discussed above, the development of antibodies or small molecules that selectively stabilize or destabilize specific multimeric GQ structures should greatly facilitate analysis of their potential biological roles in promoters.

## 6. Potential Biological Roles of RNA–DNA Hybrid G-Quadruplexes

In addition to the multimeric DNA GQs discussed so far, multimeric GQs can also form from combinations of DNA and RNA strands [124]. These structures are called DNA:RNA hybrid GQs. They can be readily assembled in vitro, and also form during transcription when guanines in the nontemplate DNA strands of genes interact with guanines in RNA molecules encoded by these genes using the second mode of multimerization (Figure 10). DNA:RNA hybrid GQs inhibit transcription, presumably by impeding the progress of RNA polymerase along the DNA template. From the perspective of gene regulation, this is a compelling mechanism because it can act as a negative feedback loop to turn transcription off when mRNA levels are sufficiently high (Figure 1A). The first hint that DNA:RNA hybrid GQs act as regulatory elements came from analysis of a region in the human mitochondrial genome called conserved sequence block II (CSB II) [125]. A G-rich element in CSB II induces transcriptional termination and promotes formation of an unusually stable structure in the template DNA. The stability of this structure suggested that it was not a conventional R-loop, in which an RNA transcript and the DNA template encoding it are held together by Watson–Crick base pairing. Instead, it was proposed that this structure was a GQ containing both DNA and RNA strands. Consistent with this hypothesis, a structure was detected by native PAGE that required for its formation the presence of both a DNA oligonucleotide corresponding to the G-rich strand of the template and an RNA oligonucleotide corresponding to the G-rich portion of the RNA transcript. Formation of this structure was inhibited when guanine bases in the DNA strand were replaced by 7-deazaguanine or adenosine, both of which normally prevent GQ formation. The circular dichroism spectrum of this structure was consistent with that of a GQ, and it was resistant to ribonucleases A and H, both of which degrade RNA strands in canonical DNA:RNA duplexes. Additional characterization using DMS fingerprinting, which can distinguish guanines in tetrads from those in single or double-stranded DNA due to their reduced sensitivity to DMS modification, and Zn-TTAPc-mediated photocleavage, which preferentially cleaves GQs, provided additional evidence that the structure is a DNA:RNA hybrid GQ [126]. Similar methods were used to show that a DNA:RNA hybrid GQ also forms in the human *NRAS* gene [127]. As was the case for the CSB II motif, the structure in the *NRAS* gene impeded T7 RNA polymerase during in vitro transcription reactions. It also inhibited expression of luciferase from plasmids transfected into cultured human cells. Only two stretches of guanines in the nontemplate strand of the gene are needed to form the DNA:RNA hybrid GQ, with the other two provided by the RNA transcript encoded by the template strand. DNA:RNA hybrid GQs containing two rather than three tetrads can also form, although they are less stable than those containing three tetrads [128]. Bioinformatic studies indicate that sequences with the potential to form DNA:RNA hybrid GQs occur in 97% of human genes [129]. These motifs are enriched in and near promoters, especially downstream of the transcription start site on the nontemplate strand, and this pattern of enrichment is evolutionarily conserved in mammals [129]. Taken together, these studies support the idea that DNA:RNA hybrid GQs represent an important regulatory element in higher eukaryotes.

Additional studies have explored the mechanism of formation of DNA:RNA hybrid GQs during transcription. Zhang and colleagues developed a method to distinguish R-loops from DNA:RNA hybrid GQs on the basis of differences in the sensitivities of these structures to ribonucleases such as RNase A [130]. This was used to show that conventional R-loops form prior to DNA:RNA hybrid GQs during in vitro transcription of a template containing the CSB II motif [130]. Formation of DNA:RNA hybrid GQs was also inhibited by the presence of a C-rich oligonucleotide complementary to the G-rich region of both the nontemplate strand and the nascent RNA transcript. Based on these experiments, it was proposed that a canonical R-loop is formed in the first round of transcription of the CSB II motif. After the RNA strand in the R-loop is displaced by RNA polymerase in the next round of transcription, it forms a DNA:RNA hybrid GQ with the nontemplate strand and prevents further transcription (Figure 10). Real-time FRET studies indicate that R-loops also stabilize DNA:RNA hybrid GQs [131], probably because they prevent the nontemplate strand from forming a duplex with the template strand. The mechanical strength of DNA:RNA hybrid GQs has also been investigated using optical tweezers [132]. These experiments used a template containing the sequence (GGGGA)_4_, which occurs downstream of the transcription start site on the nontemplate strand in several hundred human genes [132]. Transcription reactions were performed in the absence of CTP, which caused the polymerase to stall 15 nucleotides downstream of the GQ at the position of the first G in the template. Stalled complexes were tethered between beads and stretched to determine their mechanical stabilities. This showed that DNA:RNA hybrid GQs are more stable and form more readily on the nontemplate strand than DNA GQs.

Recent results from our laboratory raise the possibility that formation of DNA:RNA hybrid GQs can be regulated by biologically important small molecules [78]. These experiments used a mutant GQ previously shown by our group to bind GTP [79,133] and to form multimers [18]. When folded in the absence of GTP, this sequence forms multimeric GQs as well as monomers that are probably unfolded. When folded in the presence of physiological concentrations of GTP, however, the monomeric form of the GQ is stabilized and formation of multimers is suppressed [78]. NMR studies indicate that the GTP ligand is incorporated into a tetrad, and together with other data support a model in which GTP is incorporated into a cavity in the central tetrad of the monomer created by a G to A mutation. Hundreds of examples of sequences with the potential to form GTP-dependent structures were identified in the human genome, some of which were evolutionarily conserved in primates. Ongoing experiments in our group are investigating possible links between GTP-dependent formation of DNA:RNA hybrid GQs and transcriptional regulation.

## 7. Conclusions

GQs can form a variety of multimeric structures, ranging in size from dimers to G-wires. Most contain interfaces formed by either stacked tetrads in adjacent GQ monomers (first mode of multimerization) or guanines from two or more nucleic acid strands that hydrogen bond to form a G-tetrad (second mode of multimerization). Some progress has been made in understanding the sequence requirements of multimeric GQs. For example, overhanging nucleotides often inhibit multimerization by interfering with the stacking of tetrads, while mutations in tetrads can promote multimerization by destabilizing the monomeric form of the structure. Despite this, the existence of unusual folds such as intertwined dimers formed from sequences containing four stretches of guanines highlight our inability to predict the structures of multimeric GQs from primary sequence. The propensity of GQs to multimerize in vitro, the high concentrations of GQs in eukaryotic cells, and the advantages of multimerization as a regulatory mechanism raise the possibility that GQ multimerization could be biologically important. Although this hypothesis is only starting to be explored, several lines of evidence suggest that higher-order GQ structures form in both telomeres and promoters. Furthermore, recent studies demonstrate that DNA:RNA hybrid GQs formed between the nontemplate strands of genes and the RNA transcripts they encode inhibit transcription, and suggest that this could be an important regulatory mechanism in higher eukaryotes. We anticipate that future progress in defining the biological roles of multimeric GQs will require the development of tools analogous to those used to characterize monomeric GQs. Examples include bioinformatic models that can identify sequences with the potential to form multimeric GQs in sequenced genomes, antibodies, and small molecules specific for different types of multimeric GQs, techniques to visualize multimeric GQs in cells, and high-throughput methods to map multimeric GQs in genomic DNA (including those containing both DNA and RNA strands). The development and application of such tools has the potential to provide a wealth of new information about multimeric GQs, especially with respect to their potential biological roles.

## Figures and Tables

**Figure 1 molecules-24-03074-f001:**
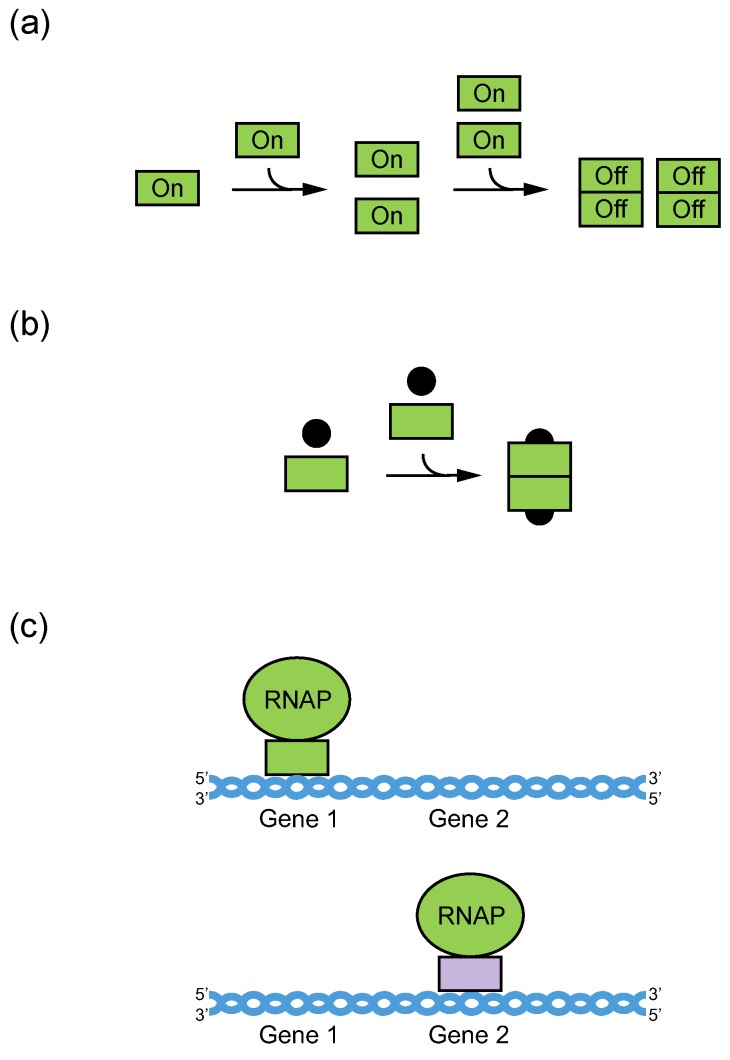
Regulation of biochemical function by multimerization. (**a**) Concentration-based control of biochemical function. In this scheme, monomers are biochemically active and dimers are not. At concentrations below the dissociation constant for dimer formation, most of the population is monomeric and in the active state, while at concentrations above the dissociation constant, most of the population is dimeric and in the inactive state. Green rectangles = nucleic acid or protein monomers. (**b**) Enhanced sensitivity to ligand concentration by cooperative binding. In this scheme, ligand binding is independent when nucleic acid or protein binding sites are monomeric but cooperative when they are linked by multimerization. This leads to all-or-none binding and enhanced sensitivity to ligand concentration. Green rectangles = nucleic acid or protein monomers; black circles = ligands. (**c**) Modulation of biochemical activity by the exchange of dimerization partners. In this scheme, the gene transcribed by RNA polymerase is determined by the DNA-binding specificity of its dimerization partner. Green ovals = RNA polymerase; green and purple rectangles = transcription factors; blue lines = DNA.

**Figure 2 molecules-24-03074-f002:**
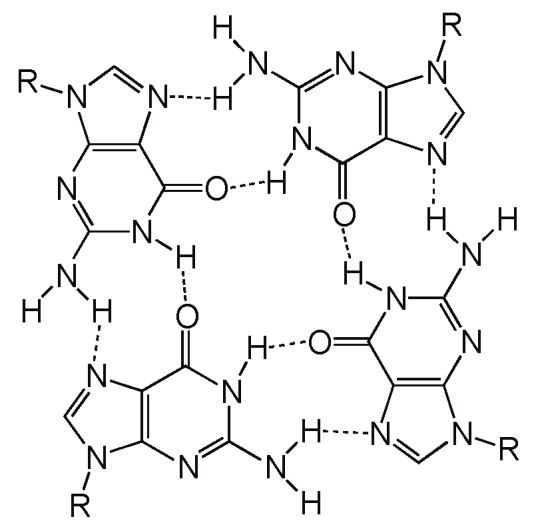
Chemical structure of a GQ (G-quadruplex) tetrad.

**Figure 3 molecules-24-03074-f003:**
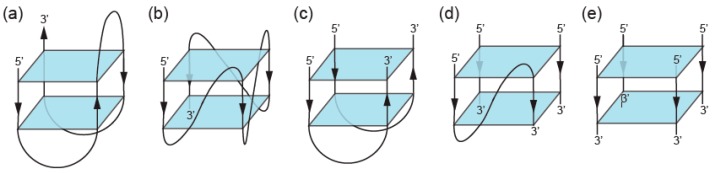
Formation of GQs from different numbers of strands. (**a**) Intramolecular (unimolecular) GQ with antiparallel strands. (**b**) Intramolecular (unimolecular) GQ with parallel strands. (**c**) Bimolecular (dimeric) GQ. (**d**) Trimolecular (trimeric) GQ. (**e**) Tetramolecular (tetrameric) GQ. Note that each of these structures can in principle contain all parallel strands, all antiparallel strands, or a mix of parallel and antiparallel strands.

**Figure 4 molecules-24-03074-f004:**
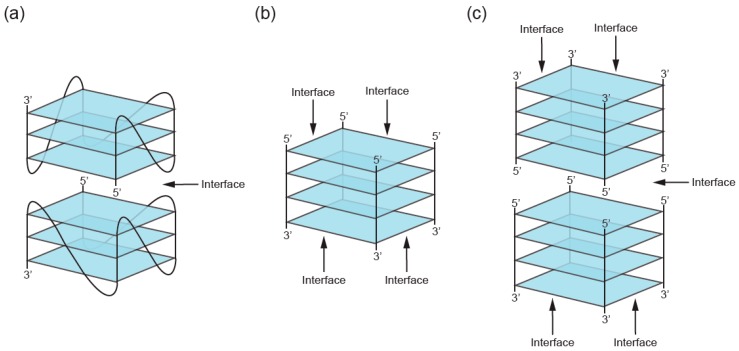
Types of interfaces in multimeric GQs. (**a**) First mode of multimerization. Interfaces are formed between tetrads which stack on top of one another in a 5′ to 5′, 3′ to 3′, or 5′ to 3′ arrangement. (**b**) Second mode of multimerization. Interfaces are formed within tetrads made up of guanines from multiple DNA strands. (**c**) Structure combining these two modes of multimerization.

**Figure 5 molecules-24-03074-f005:**
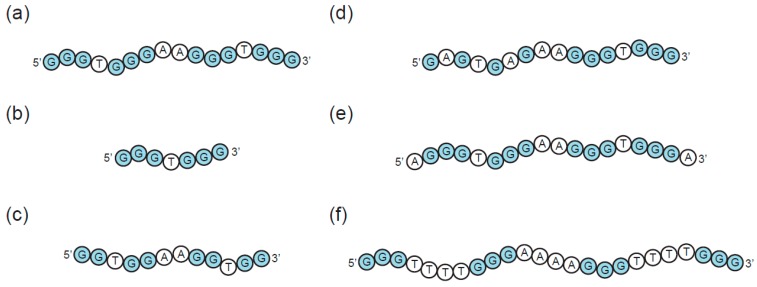
Sequence requirements of multimeric GQs. (**a**) Example of a canonical GQ. (**b**) Variant containing two rather than four G-runs. Such a sequence can form a multimeric GQ but not a monomeric one. (**c**) Variant containing G-runs of two rather than three nucleotides. In some cases, such sequences form multimeric rather than monomeric structures. (**d**) Variant containing mutations in tetrads. Such mutations can induce formation of dimeric and tetrameric structures. (**e**) Variant containing overhanging nucleotides. Such variants typically cannot stack via the first mode of multimerization, but the ability to interact via the second mode of multimerization is unaffected. (**f**) Variant containing extended loops. Longer loops favor formation of antiparallel rather than parallel GQs, and such loops can interfere with the ability of GQs to stack via the first mode of multimerization.

**Figure 6 molecules-24-03074-f006:**
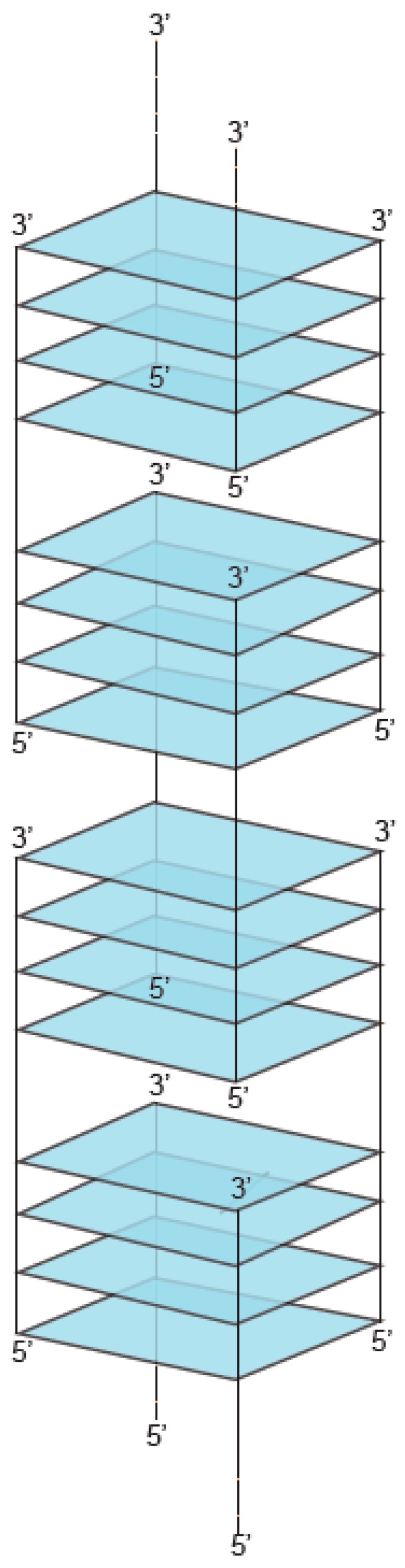
Structure of a G-wire.

**Figure 7 molecules-24-03074-f007:**
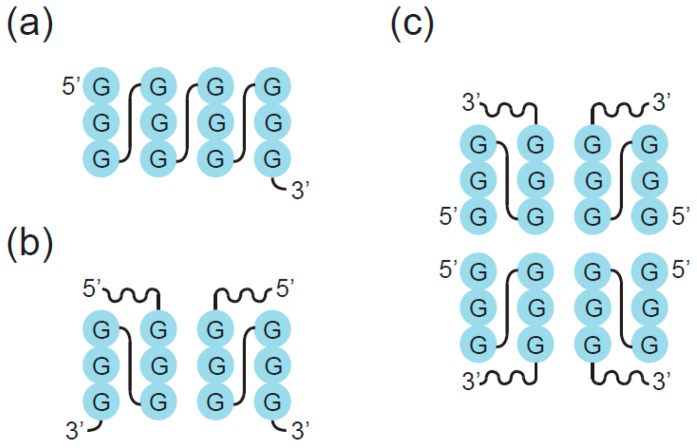
Mutations in tetrads can induce GQ multimerization. (**a**) Secondary structure of a GQ we are studying in our group with the sequence GGGTGGGAAGGGTGGGA. We previously generated a library containing all possible mutations in the central tetrad in this structure (at positions 2, 6, 11, and 15) and tested these variants for a series of biochemical activities associated with GQs [18,77,78,79]. (**b**) Proposed secondary structure of dimers formed by variants containing mutations at positions 2, 6, or both in the central tetrad of the reference GQ. The 5′ part of this structure, which contains the mutated nucleotides, is represented by a wavy black line. (**c**) Proposed secondary structure of tetramers formed by variants containing mutations at positions 11, 15, or both in the central tetrad of the reference GQ. The 3′ part of this structure, which contains the mutated nucleotides, is represented by a wavy black line.

**Figure 8 molecules-24-03074-f008:**
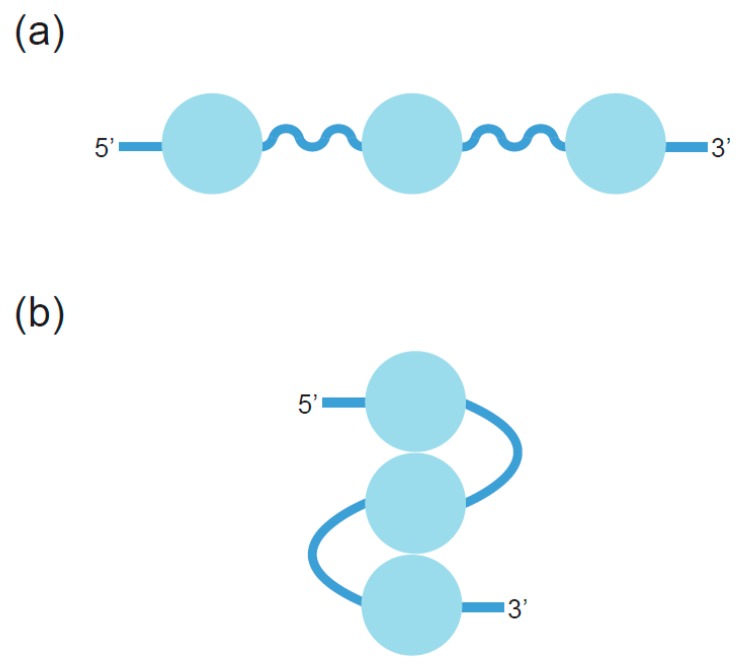
Possible structures of telomeric GQs. (**a**) Beads-on-a-string model in which telomeric GQs do not interact. (**b**) Model in which telomeric GQs stack on one another to form higher-order structures. Blue circles represent GQ structures.

**Figure 9 molecules-24-03074-f009:**
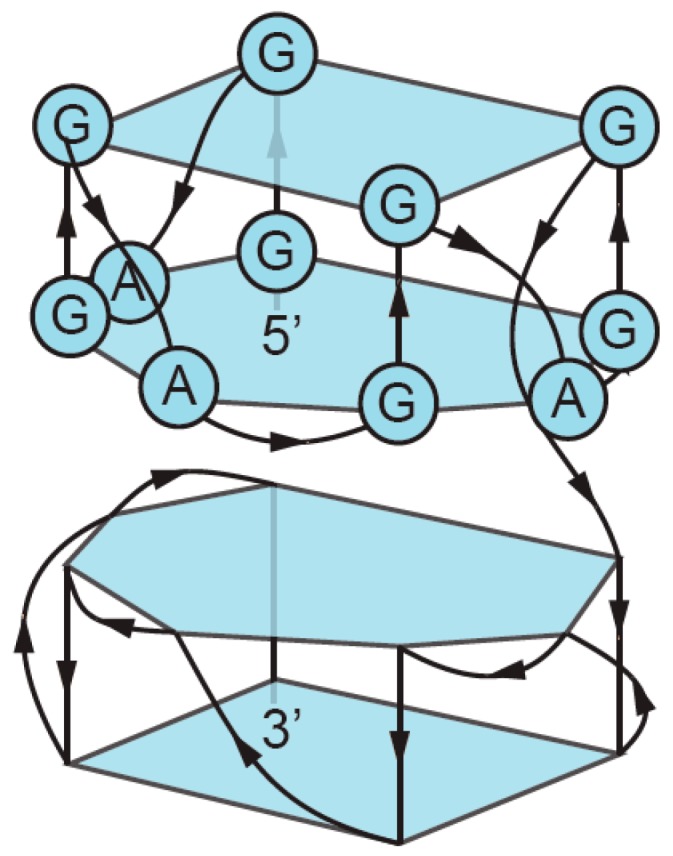
Structure of a dimeric GQ formed by the sequence (GGA)_8_. See [64] and [107] for more information about this structure.

**Figure 10 molecules-24-03074-f010:**
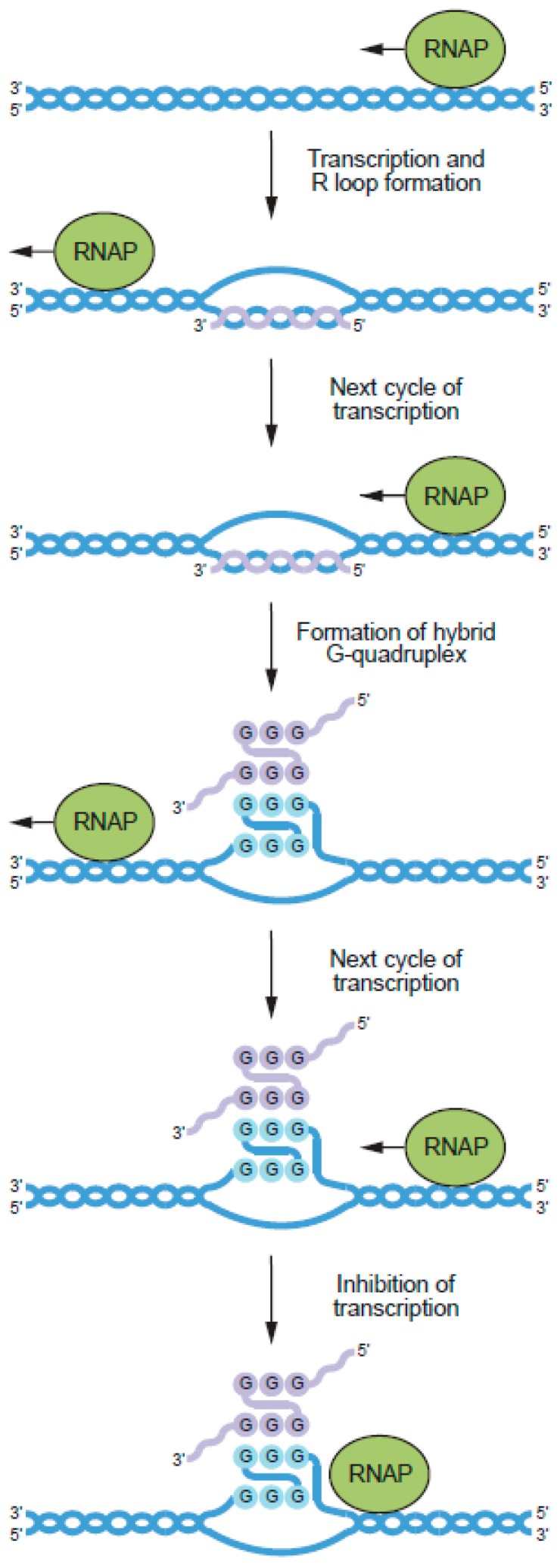
Model for the regulation of transcription by DNA:RNA hybrid GQs, formed between the noncoding strands of G-rich genes and RNA molecules transcribed from these genes. The newly synthesized RNA transcript is shown in purple. See [130] for more information about this model.

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
