# Peer review of "Structure and Function of Multimeric G-Quadruplexes"

_molecules, 2019, doi:10.3390/molecules24173074_

Round 1

Reviewer 1 Report

The authors provided an extensive review on the structure of G4 and on the possibility of these structures to form multimers.

The paper is clear and well written. I’d like to express only a few comments:

The authors should stress that in chapters 2 and 3 the described data are recorded only in vitro. Figure 1 is not very clear, the green blocks are misleading if G4s are intended, please provide a clear-cut image. In the introduction and in chapter 4 the authors mention only human DNA G4s. In the recent years many works focused also on the presence of multiple G4 also at the pathogen level (i.e. viruses). For viruses several works have been published that described the potential presence of multiple G4s and their regulation of viral promoters (https://doi.org/10.1371/journal.pcbi.1006675; Molecules 2019, 24(13), 2375; https://doi.org/10.3390/molecules24132375). Please add a brief section on this point.

Author Response

The authors provided an extensive review on the structure of G4 and on the possibility of these structures to form multimers.

The paper is clear and well written. I’d like to express only a few comments:

The authors should stress that in chapters 2 and 3 the described data are recorded only in vitro.

We have emphasized this point by adding the sentence "the types of sequences that can adopt GQ structures in vitro have been extensively characterized" to Chapter 2 and the phrase "in vitro" to the first sentence of Chapter 3.

Figure 1 is not very clear, the green blocks are misleading if G4s are intended, please provide a clear-cut image.

We have expanded the legend to more clearly explain Figure 1.

In the introduction and in chapter 4 the authors mention only human DNA G4s. In the recent years many works focused also on the presence of multiple G4 also at the pathogen level (i.e. viruses). For viruses several works have been published that described the potential presence of multiple G4s and their regulation of viral promoters (https://doi.org/10.1371/journal.pcbi.1006675; Molecules 2019, 24(13), 2375; https://doi.org/10.3390/molecules24132375). Please add a brief section on this point.

We have cited these papers and added the sentence "In addition, several viral promoters containing multiple sequences with the potential to form GQs have recently been reported."

Reviewer 2 Report

This review was interesting. I recommand this publication after minor revision.

L. 110: The sentence must be deleted and the reference added in the previous one.

L. 129: "Sen and Gilbert, 1988" must be removed.

L. 135, L. 162, L. 225, L; 302. L. 358: delete the word wrap

L. 267: a space is mising after :

L. 308: what is EM ?

Figure 8: what is the blue ball ? A explanation must be added in te caption.

Author Response

This review was interesting. I recommand this publication after minor revision.

110: The sentence must be deleted and the reference added in the previous one.

We have made this change.

129: "Sen and Gilbert, 1988" must be removed.

We have made this change.

135, L. 162, L. 225, L; 302. L. 358: delete the word wrap

We have made this change.

267: a space is mising after :

We have changed the colon to a dash.

308: what is EM ?

We have changed "AFM and EM" to "atomic force and electron microscopy."

Figure 8: what is the blue ball ? A explanation must be added in te caption.

The blue circle is a G-quadruplex. We have indicated this in the figure legend.

Reviewer 3 Report

Sofia Kolesnikova and Edward A. Curtis present in a paper named “Structure and function of multimeric G-quadruplexes” an exhaustive review of intermolecular quadruplexes and various modes of the interactions of involved strands. They also briefly categorize the multimolecular G-quadruplexes, including terminology. The paper is well written and readable and an appropriate number of references supports most of the statements.

Taking into account, the authors give significant portion of space to the arrangement of monomer units within long telomere sequences, i.e. within one molecule, they should also consider brief mentioning the parallel-quadruplex associates within one strand, presented by B. Kankia (Kankia 2014; Kankia 2018), and possibly also the left-handed quadruplexes presented by group of A.T. Phan (Chung et al., 2015; Bakalar et al., 2019) that, in fact, represent a stacked dimers of two two-tetrad parallel quadruplexes.

l. 170 - In figure 5f and the corresponding text, the authors show the extension of just one, central loop with description “Longer loops favor formation of antiparallel rather than parallel GQs and such loops can interfere with the ability of GQs to stack via the first mode of multimerization”. This is fine for illustration but might be a bit misleading, as the long loop sequences do not prevent the formation of propeller loop per se (Yue et al., 2011). The remaining two single nucleotide loops ensure the formation of parallel fold (l. 259), though it is not clear whether such sequence will multimerize via the first mode. The sequence will probably not form an antiparallel quadruplex.

l. 220 – The authors claim that “Despite containing an exposed 5’ tetrad, only monomers were observed when this sequence was analyzed on native gels.” In this context, I think it should be mentioned that the sequences used for gels were 5’-radiolabelled, i.e. with 5’ end phosphate group. As the authors mention later (l. 247), “In some cases even the introduction of a 5’ phosphate group is sufficient to inhibit 5’-5’ stacking [73]”.

l. 320 – “…the antibody has similar affinities for both types of structures.” Could you provide a reference for this statement?

In conclusion, the manuscript is well written and brings very useful overview of multimeric G-quadruplexes. I would recommend the manuscript for publication in the Molecules.

Author Response

Sofia Kolesnikova and Edward A. Curtis present in a paper named “Structure and function of multimeric G-quadruplexes” an exhaustive review of intermolecular quadruplexes and various modes of the interactions of involved strands. They also briefly categorize the multimolecular G-quadruplexes, including terminology. The paper is well written and readable and an appropriate number of references supports most of the statements.

Taking into account, the authors give significant portion of space to the arrangement of monomer units within long telomere sequences, i.e. within one molecule, they should also consider brief mentioning the parallel-quadruplex associates within one strand, presented by B. Kankia (Kankia 2014; Kankia 2018), and possibly also the left-handed quadruplexes presented by group of A.T. Phan (Chung et al., 2015; Bakalar et al., 2019) that, in fact, represent a stacked dimers of two two-tetrad parallel quadruplexes.

We have added the sentence "a similar arrangement of stacked monomers has recently been described in the context of non-telomeric oligonucleotides containing multiple GQs" and cited the two Kankia papers.

170 - In figure 5f and the corresponding text, the authors show the extension of just one, central loop with description “Longer loops favor formation of antiparallel rather than parallel GQs and such loops can interfere with the ability of GQs to stack via the first mode of multimerization”. This is fine for illustration but might be a bit misleading, as the long loop sequences do not prevent the formation of propeller loop per se (Yue et al., 2011). The remaining two single nucleotide loops ensure the formation of parallel fold (l. 259), though it is not clear whether such sequence will multimerize via the first mode. The sequence will probably not form an antiparallel quadruplex.

To address this point we have extended each of the loops in the sequence shown in Figure 5F.

220 – The authors claim that “Despite containing an exposed 5’ tetrad, only monomers were observed when this sequence was analyzed on native gels.” In this context, I think it should be mentioned that the sequences used for gels were 5’-radiolabelled, i.e. with 5’ end phosphate group. As the authors mention later (l. 247), “In some cases even the introduction of a 5’ phosphate group is sufficient to inhibit 5’-5’ stacking [73]”.

A control experiment described in Supplementary Figure 1 of reference 18 shows that this sequence (labeled "GGGG" in Supplementary Figure 1) also forms monomers when it contains a 5' hydroxyl group rather that a 5' phosphate. We have clarified this point by replacing the sentence "despite containing an exposed 5' tetrad, only monomers were observed when this sequence was analyzed on native gels." with the sentence "despite containing an exposed 5' tetrad, only monomers were observed when versions of this sequence which contained either a 5' hydroxyl or a 5' phosphate group were analyzed on native gels."

320 – “…the antibody has similar affinities for both types of structures.” Could you provide a reference for this statement?

We have removed this phrase.

In conclusion, the manuscript is well written and brings very useful overview of multimeric G-quadruplexes. I would recommend the manuscript for publication in the Molecules